# Fault Detection Based on Multi-Dimensional KDE and Jensen–Shannon Divergence

**DOI:** 10.3390/e23030266

**Published:** 2021-02-24

**Authors:** Juhui Wei, Zhangming He, Jiongqi Wang, Dayi Wang, Xuanying Zhou

**Affiliations:** 1College of Liberal Arts and Sciences, National University of Defense Technology, Changsha 410073, China; weijuhui_nudt@nudt.edu.cn (J.W.); wangjq@nudt.edu.cn (J.W.); zhouxy@nudt.edu.cn (X.Z.); 2Beijing Institute of Spacecraft System Engineering, China Academy of Space Technology, Beijing 100094, China; dayiwang@163.com

**Keywords:** fault detection, optimal bandwidth, kernel density estimation, JS divergence, bearing

## Abstract

Weak fault signals, high coupling data, and unknown faults commonly exist in fault diagnosis systems, causing low detection and identification performance of fault diagnosis methods based on $T^{2}$ statistics or cross entropy. This paper proposes a new fault diagnosis method based on optimal bandwidth kernel density estimation (KDE) and Jensen–Shannon (JS) divergence distribution for improved fault detection performance. KDE addresses weak signal and coupling fault detection, and JS divergence addresses unknown fault detection. Firstly, the formula and algorithm of the optimal bandwidth of multidimensional KDE are presented, and the convergence of the algorithm is proved. Secondly, the difference in JS divergence between the data is obtained based on the optimal KDE and used for fault detection. Finally, the fault diagnosis experiment based on the bearing data from Case Western Reserve University Bearing Data Center is conducted. The results show that for known faults, the proposed method has 10% and 2% higher detection rate than $T^{2}$ statistics and the cross entropy method, respectively. For unknown faults, $T^{2}$ statistics cannot effectively detect faults, and the proposed method has approximately 15% higher detection rate than the cross entropy method. Thus, the proposed method can effectively improve the fault detection rate.

## 1. Introduction

The development of industrial informatization has given rise to a large amount of data in various fields. This has led to data processing becoming a difficult problem in the industry, especially for fault diagnosis. The explosive growth of data provides more information, and therefore, typical data analysis theories often fail in achieving the necessary results. The main reason for this failure can be attributed to the typical data analysis theory that often sets the data distribution type through prior information and performs analyses based on this assumption. Once the distribution type is set, the subsequent analysis can perform the estimation and parametric analysis based on only that distribution type; however, with the growth of data, more information is provided, and thus, the type of data distribution will need to be modified. As a nonparametric estimation method, kernel density estimation (KDE) is the most suitable method for the massive amount of the current data. KDE does not employ a priori assumption for the overall data distribution, and it directly starts from the sample data. When the sample size is sufficient, the KDE can approximate different distributions. Furthermore, Sheather and Jones [1] provides the optimal bandwidth estimation formula for a one-dimensional KDE and proves that the kernel function is asymptotically unbiased and consistent in the density estimation. However, with the growth of the dimension, the multidimensional KDE becomes more complex, and its optimal bandwidth formula is not provided. The distribution of multidimensional data has been described to a certain extent by estimating the kernel density of the reduced data in different dimensions Muir [2], Laurent [3]. In fact, the optimal KDE of multidimensional data is a problem that needs to be studied further.

In the field of fault diagnosis, an essential problem is measuring the difference between samples. A frequency histogram has been used to indicate the distribution difference between two samples Sugumaran and Ramachandran [4], Scott [5]; however, there are three shortcomings to this method: (1) the large number of discrete operations require a higher amount of time; (2) the process depends on the selection of the interval, which is more subjective; (3) there is no intuitive index to reflect this difference. In fact, based on KDE, the JS divergence can be used to measure the difference in data distribution, which can overcome the above shortcomings to a certain extent. For example, the failure of a rolling bearing, which is a key component of mechanical equipment, will have a serious effect on the safe and stable operation of the equipment, and the incipient fault detection of rolling bearings can help avoid equipment running with faults and avoid causing serious safety accidents and economic losses, which has important practical and engineering significance.

In Saruhan et al. [6], vibration analysis of rolling element bearings (REBs) defects is studied. The REBs are the most widely used mechanical parts in rotating machinery under high load and high rotational speeds. In addition, characteristics of bearing faults are analyzed in detail in references Razavi-Far et al. [7], Harmouche et al. [8]. Compared with traditional fault diagnosis, the fault diagnosis of rolling bearings is more complex:The fault signal is weak: Bearing data is a type of high-frequency data, and the fault signal is often covered by these high-frequency signals, thereby leading to the failure of traditional fault diagnosis methods. KDE is highly accurate in describing data distribution, so it can identify weak signals.Data is highly coupled: Bearing data is reflected in the form of a vibration signal, and there is strong coupling in different dimension signals, thereby making fault diagnosis difficult. Multi-dimensional KDE plays an important role in depicting the correlation of data, which can characterize the relationship between different dimensions of data.Incomplete data set: Most bearings work under normal conditions, and the fault data collected are often fewer, which makes the data incomplete, thereby resulting in the imperfection of the fault data set and increasing the difficulty of fault detection. The fault detection method constructed by JS divergence can deal with unknown faults and incomplete data sets without using additional data sets.

To overcome these problems, in-depth research has been conducted on this topic. Fault detection technology based on trend elimination and noise reduction has been proposed previously He et al. [9], Demetriou and Polycarpou [10]. The signal trend ratio is enhanced by eliminating the trend, and the signal–noise ratio is enhanced by noise reduction, and therefore, the fault detection effect is improved. However, this method uses the traditional detection method and cannot effectively solve the problem of data coupling. In reference Zhang et al. [11], Fu et al. [12], a fault detection method based on PCA dimension reduction and modal decomposition feature extraction is proposed. For multidimensional data, PCA dimension reduction is performed to reduce data dimensions and eliminate correlation between different dimensions. Then, the modal decomposition method is used to extract features among dimensions for fault detection. This method can effectively solve the strong coupling between data; however, it will lose some information in the process of PCA dimension reduction, and it leads to a reduction in the fault detection effect. In reference Itani et al. [13], Kong et al. [14], Jones and Sheather [15], Desforges et al. [16], a bearing fault detection method based on KDE is proposed. These studies analyzed the feasibility of KDE method in fault detection, and combined different classification methods for experiments. However, these methods only use one-dimensional KDE, and cannot directly describe high-dimensional data.

The data distribution is reconstructed by KDE and the cross-entropy function is constructed to measure the distribution difference for improving the fault detection results. However, this method cannot reflect the correlation between different dimensions, and the cross-entropy function is not precise in the description of density distribution, which leads to a reduction in the fault detection effect, especially for unknown fault detection, which is not included in the fault set.

In this study, the KDE method is extended to multidimensional data to avoid information loss caused by the KDE for each dimension, and to better describe the density probability distribution of the data. Meanwhile, this study improves the traditional method using the cross-entropy function as the measurement of density distribution difference, and it uses JS divergence as the measurement of density distribution difference, thereby avoiding the relativity caused by the cross-entropy function. Most fault identification methods are based only on distance measurement; however, only relying on distance measurement cannot effectively detect unknown faults. Based on JS divergence, distribution characteristics of JS divergence between the sample density distribution and population density distribution are derived using the sliding window principle. Thus, the detection threshold of fault identification is assigned to realize the identification of unknown faults.

This paper is based on the following structure. In Section 2, the trend elimination method and detection method are introduced, and the intrinsic and extrinsic signals in the observation data are separated. Then, the fault detection threshold is constructed via statistics. In Section 3, the KDE method is extended to multidimensional data, and the optimal bandwidth is derived. Then, JS divergence is employed to measure the difference between probability distributions of different densities. In Section 4, the sliding window principle is used to sample the training data to obtain the distribution characteristics of JS divergence between the sample density distribution and the overall density distribution, and the detection threshold of fault identification is obtained using the KDE method. In Section 5, the normal data, two known faults, and one unknown fault are identified using the bearing data of the Case Western Reserve University Bearing Data Center as the fault diagnosis data. The experimental results show that the method can identify all types of faults well.

## 2. $T^{2}$ Statistics Fault Detection

In the operation process of the complex equipment or systems, the common observation state can be divided into intrinsic and extrinsic parts. In general, the intrinsic part represents the main working state of the system, which has a certain trend, monotony, and periodicity. The extrinsic part represents system noise, which has a certain zero mean value, high frequency vibration, and statistical stability. For the intrinsic part, the state equation of the system can be used to describe the law. When a fault occurs in the intrinsic part, the symptoms are relatively significant, and the corresponding fault detection methods are relatively mature. However, for high-frequency vibration signals, the incipient fault is often hidden in the extrinsic part, which is easily covered by noise. Therefore, it is necessary to analyze the observed data in depth.

### 2.1. Signal Decomposition

In the initial operation stage of the equipment, the unstable operation of the system causes large data fluctuations, which will not only have a great effect on the system trend, but also affect the statistical characteristics of the data. Therefore, it is necessary to truncate the data to remove unstable signals [9]. The corresponding time of the time series after removing the nonstationary period data is $t_{1},t_{2},\ldots ,t_{m}$, and the following *m* observation data are obtained:(1)$$Y\;=\;\left[y\left(t_{1}\right),y\left(t_{2}\right),\ldots ,y\left(t_{m}\right)\right].$$

Each sampling $y\left(t_{i}\right)$ contains *n* features, which are expressed as components in the form of
(2)$$y\left(t_{i}\right)={\left[y_{1}\left(t_{i}\right),y_{2}\left(t_{i}\right),\ldots ,y_{n}\left(t_{i}\right)\right]}^{T},i=1,2,\ldots ,m.$$

Then, the data Y can be decomposed into
(3)$$Y=\hat{Y}+R,$$
where $\hat{Y}$ denotes the intrinsic part, which is composed of trend, and R denotes the extrinsic part, which is composed of observation noise and fault data.

The intrinsic part is composed of multiple signals. Selecting the appropriate basis function $f\left(t\right)={\left[f_{1}\left(t\right),f_{2}\left(t\right),\ldots ,f_{s}\left(t\right)\right]}^{T}$ can help describe the intrinsic part. By traversing *m* data to model the nonlinear data Y,
(4)$$\left[y_{1},y_{2},\ldots ,y_{m}\right]=\left[\begin{array}{cccc} \beta_{1}^{\left(1\right)} & \beta_{2}^{\left(1\right)} & \ldots & \beta_{s}^{\left(1\right)}\\ \beta_{1}^{\left(2\right)} & \beta_{2}^{\left(2\right)} & \ldots & \beta_{s}^{\left(2\right)}\\ \vdots & \vdots & \ddots & \vdots \\ \beta_{1}^{\left(n\right)} & \beta_{2}^{\left(n\right)} & \ldots & \beta_{s}^{\left(n\right)} \end{array}\right]\left[\begin{array}{cccc} f_{0}\left(t_{1}\right) & f_{0}\left(t_{2}\right) & \ldots & f_{0}\left(t_{m}\right)\\ f_{1}\left(t_{1}\right) & f_{1}\left(t_{2}\right) & \ldots & f_{1}\left(t_{m}\right)\\ \vdots & \vdots & \ddots & \vdots \\ f_{s}\left(t_{1}\right) & f_{s}\left(t_{2}\right) & \ldots & f_{s}\left(t_{m}\right) \end{array}\right].$$

Note that
(5)$$F\overset{\Delta }{\;=}\left[\begin{array}{cccc} f_{0}\left(t_{1}\right) & f_{0}\left(t_{2}\right) & \ldots & f_{0}\left(t_{m}\right)\\ f_{1}\left(t_{1}\right) & f_{1}\left(t_{2}\right) & \ldots & f_{1}\left(t_{m}\right)\\ \vdots & \vdots & \ddots & \vdots \\ f_{s}\left(t_{1}\right) & f_{s}\left(t_{2}\right) & \ldots & f_{s}\left(t_{m}\right) \end{array}\right],\beta \overset{\Delta }{\;=}\left[\begin{array}{cccc} \beta_{1}^{\left(1\right)} & \beta_{2}^{\left(1\right)} & \ldots & \beta_{s}^{\left(1\right)}\\ \beta_{1}^{\left(2\right)} & \beta_{2}^{\left(2\right)} & \ldots & \beta_{s}^{\left(2\right)}\\ \vdots & \vdots & \ddots & \vdots \\ \beta_{1}^{\left(n\right)} & \beta_{2}^{\left(n\right)} & \ldots & \beta_{s}^{\left(n\right)} \end{array}\right]$$

Then, Equation (Equation 4) can be expressed as
(6)Y=βF.

Thus, the efficient estimator of β is
(7)$$\hat{\beta }=YF^{T}{\left(FF^{T}\right)}^{-1}.$$

Using Equations (Equation 3) and (Equation 7), the signal can be decomposed into
(8)$$\left\{\begin{array}{cc} \; & \hat{Y}=\hat{\beta }F=YF^{T}{\left(FF^{T}\right)}^{-1}F\\ \; & R=Y-\hat{Y}=Y\left(I-F^{T}{\left(FF^{T}\right)}^{-1}F\right) \end{array}\right.$$

Usually, the choice of the basis function is a problem worthy of discussion, and it depends on prior knowledge of practical application scenarios; however, this is not the focus of this paper, and is therefore not covered here.

**Remark** **1.**
*For the bearing data, the data is generally stable and periodic. Therefore, Fourier transform is usually used to extract periodic features instead of more complex basis functions, such as a polynomial basis function and wavelet basis function.*


### 2.2. $T^{2}$ Statistics Detection

For simplicity, remember $r_{i}=r\left(t_{i}\right),i=1,2,\ldots ,m$. According to Equation (Equation 8), the training data after signal decomposition are $R=\left[r_{1},r_{2},\ldots ,r_{m}\right]$, which is generally considered a normal random vector with expectation of 0, so that
(9)$$r_{i}∼N\left(0,\Sigma \right),$$
where Σ denotes the total covariance matrix. When the covariance matrix Σ is unknown, the unbiased estimation is given by
(10)$$\hat{\Sigma }=\frac{RR^{T}}{m-1}.$$

Let $Z=\left[z_{1},z_{2},\ldots ,z_{p}\right]$ be the data in the test window to be tested; the sample mean value $\bar{z}$ is
(11)$$\bar{z}=\frac{1}{p}\sum_{i=1}^{p}z_{i}.$$

Then, $\bar{z}$ is still normal distributed and
(12)$$\bar{z}∼N\left(0,\frac{1}{p}\Sigma \right).$$

The $T^{2}$ statistics can be constructed as
(13)$$T^{2}=p{\bar{z}}^{T}{\hat{\Sigma }}^{-1}\bar{z}.$$

Reference Solomons and Hotelling [17] reports that the distribution of the $T^{2}$ statistic satisfies
(14)$$\frac{m-n}{n\left(m-1\right)}T^{2}=\frac{p\left(m-n\right)}{n\left(m-1\right)}{\bar{z}}^{T}{\hat{\Sigma }}^{-1}\bar{z}∼F\left(n,m-n\right).$$

Therefore, if the significance level is α, we can get that
(15)$$\frac{m-n}{n\left(m-1\right)}T^{2}=\frac{l\left(m-n\right)}{n\left(m-1\right)}{\bar{z}}^{T}{\hat{\Sigma }}^{-1}\bar{z}<F_{\alpha }\left(n,m-n\right).$$

The testing data Z and the training data R both come from the same mode; otherwise, they are considered different. The error rate of this criterion is α.

## 3. Optimal Kernel Density Estimation

Section 2 introduces the fault detection method based on $T^{2}$ statistics, including the signal decomposition technology and fault detection method based on the $T^{2}$ statistics. However, the fault detection method based on the $T^{2}$ statistics assumes that data satisfies the normal distribution, while the actual observation data may not meet the hypothesis, which can lead the discriminant performance of the $T^{2}$ statistics to not satisfy the design requirements. In addition, the statistics test the data from the angle of the intrinsic part $\hat{Y}$ and covariance matrix $\hat{\Sigma }$. These two attributes are not sufficient to describe all statistical characteristics of the system. When the incipient fault is submerged by data noise, it is easy to miss the detection. In this study, a KDE method for multidimensional data is constructed to describe the probability and statistical characteristics of the data more accurately.

### 3.1. Optimal Bandwidth Theorem

For the observed data, the frequency histogram can be used to show its statistical characteristics directly. However, in the actual application process, the frequency histogram is a discrete statistical method, the interval number of the histogram is difficult to divide, and more importantly, the discretization operation inconveniences the subsequent data processing. To overcome these limitations, the KDE method is proposed. This method is a nonparametric estimation method that estimates the population probability density distribution directly by sampling data.

For any point $x\in R^{n}$, assuming that the probability density of a certain mode is $f\left(x\right)$, the kernel density of $f\left(x\right)$ is estimated based on the sampling data $R=\left[r_{1},r_{2},\ldots ,r_{m}\right]$ in Section 2.1. As reported in reference Rao [18], the estimation formula is
(16)$${\hat{f}}_{K}\left(x,h_{m}\right)=\frac{1}{mh_{m}^{n}}\sum_{i=1}^{m}K\left(\frac{r_{i}-x}{h_{m}}\right),$$
where m,
n,
$K\left(\cdot \right),$ and $h_{m}$ denote the number of sampling data, dimension of sampling data, kernel function, and bandwidth, respectively.

For the sake of convenience in the following discussions, in the case of no doubt,
(17)$$\left\{\begin{array}{cc} \; & {\hat{f}}_{K}\left(x\right)≜{\hat{f}}_{K}\left(x,h_{m}\right)\\ \; & \int g\left(x\right)dx≜\int_{x\in R^{n}}g\left(x\right)dx \end{array}\right.$$

The kernel function $K\left(\cdot \right)$ satisfies $\int K\left(x\right)dx=1$; therefore, $\int K\left(\frac{r_{i}-x}{h_{m}}\right)dx=h_{m}^{n}$, that is, $\int {\hat{f}}_{K}\left(x\right)dx=1$. Thus, ${\hat{f}}_{K}\left(x\right)$ satisfies both positive definiteness, continuity, and normality. Therefore, it is reasonable to use it as the KDE. The Gaussian kernel function is a good choice as given by
(18)Kx=2π−nn22e−xTxxTx22

In this study, the performance of the kernel density estimator is characterized by the mean integral square error (MISE).
(19)$$\mathrm{MISE}\left({\hat{f}}_{K}\left(x\right)\right)=\int E{\left[{\hat{f}}_{K}\left(x\right)-f\left(x\right)\right]}^{2}dx$$

Reference Rao [18] shows that the estimation result ${\hat{f}}_{K}\left(x\right)$ is not sensitive to the selection of the kernel function $K\left(\cdot \right)$; that is, the MISE of the estimation results obtained using different kernel functions is almost the same, which is reflected in the subsequent derivation process. In addition, the MISE depends on the selection of the bandwidth $h_{m}$. If $h_{m}$ is too small, the density estimation ${\hat{f}}_{K}\left(x\right)$ shows an irregular shape because of the increase in the randomness. While $h_{m}$ is too large, density estimation ${\hat{f}}_{K}\left(x\right)$ is too averaged to show sufficient detail.

The optimal bandwidth formula is provided in the following theorem, and it is one of the key theoretical results of this study.

**Theorem** **1.**
*For any dimensional probability density function $f\left(\cdot \right)$ and any kernel function $K\left(\cdot \right)$ with a symmetric form, if ${\hat{f}}_{K}\left(\cdot \right)$ in Equation (Equation 16) is used to estimate $f\left(\cdot \right)$, and if the function $\mathrm{tr}\left(\frac{\partial^{2}f\left(x\right)}{\partial x\partial x^{T}}\right)$ with respect to x is integrable when the $\mathrm{MISE}\left({\hat{f}}_{K}\left(\cdot \right)\right)$ in Equation (Equation 19) is the minimum, the bandwidth $h_{m}$ satisfies*
(20)hm=mdK2n3cK∫tr∂2fx∂x∂xT2dx−11n+4n+4,

*where $c_{K}$ and $d_{K}$ are two constant values given by*
(21)$$\left\{\begin{array}{cc} \; & c_{K}=\int K^{2}\left(x\right)dx\\ \; & d_{K}=\int x^{T}xK^{2}\left(x\right)dx \end{array}\right.$$

*Equation (Equation 20) is called the optimal bandwidth formula and $h_{m}$ denotes the optimal bandwidth.*


A detailed proof of this theorem is given below.

**Proof.** It can be proved that the following two equations hold
(22)$$\left\{\begin{array}{cc} \; & E\left[{\hat{f}}_{K}\left(x\right)\right]=\int K\left(u\right)f\left(x+h_{m}u\right)du\\ \; & E\left[{\hat{f}}_{K}^{2}\left(x\right)\right]=\frac{\int K^{2}\left(u\right)f\left(x+h_{m}u\right)du}{mh_{m}^{n}}+\frac{\left(m-1\right){\left(\int K\left(u\right)f\left(x+h_{m}u\right)du\right)}^{2}}{m} \end{array}\right.$$In fact,
(23)$$\begin{array}{cc} E\left[{\hat{f}}_{K}\left(x\right)\right] & =\int \ldots \int \prod_{i=1}^{m}f\left(r_{i}\right)\frac{1}{mh_{m}^{n}}\sum_{i=1}^{m}K\left(\frac{r_{i}-x}{h_{m}}\right)dr_{m}\ldots dr_{1}\\ & =\frac{1}{mh_{m}^{n}}\sum_{i=1}^{m}\int f\left(r\right)K\left(\frac{r-x}{h_{m}}\right)dr\\ & =\int f\left(x+h_{m}u\right)K\left(u\right)du. \end{array}$$In addition,
(24)$$\begin{array}{c} E\left[{\hat{f}}_{K}^{2}\left(x\right)\right]=\int \ldots \int \prod_{i=1}^{m}f\left(r_{i}\right){\left({\left(mh_{m}^{n}\right)}^{-1}\sum_{i=1}^{m}f\left(r_{i}\right)K\left(\frac{r_{i}-x}{h_{m}}\right)\right)}^{2}dr_{1}\ldots dr_{m}\\ ={\left(mh_{m}^{n}\right)}^{-2}\int \ldots \int \prod_{i=1}^{m}f\left(r_{i}\right){\left(\sum_{i=1}^{m}f\left(r_{i}\right)K\left(\frac{r_{i}-x}{h_{m}}\right)\right)}^{2}dr_{1}\ldots dr_{m}\\ ={\left(mh_{m}^{n}\right)}^{-2}\int \ldots \int \prod_{i=1}^{m}f\left(r_{i}\right)\left(\sum_{i=1}^{m}K^{2}\left(\frac{r_{i}-x}{h_{m}}\right)+\sum_{i\neq j}^{m}K\left(\frac{r_{i}-x}{h_{m}}\right)K\left(\frac{r_{j}-x}{h_{m}}\right)\right)dr_{1}\ldots dr_{m}\\ ={\left(mh_{m}^{n}\right)}^{-2}\int \ldots \int \left(\prod_{i=1}^{m}f\left(r_{i}\right)\sum_{i=1}^{m}K^{2}\left(\frac{r_{i}-x}{h_{m}}\right)+\prod_{i=1}^{m}f\left(r_{i}\right)\sum_{i\neq j}^{m}K\left(\frac{r_{i}-x}{h_{m}}\right)K\left(\frac{r_{j}-x}{h_{m}}\right)\right)dr_{1}\ldots dr_{m}\\ ={\left(mh_{m}^{n}\right)}^{-2}\left(\sum_{i=1}^{m}\int f\left(r_{i}\right)K^{2}\left(\frac{r_{i}-x}{h_{m}}\right)dr+\sum_{i\neq j}^{m}\int \int \left(f\left(r_{i}\right)K\left(\frac{r_{i}-x}{h_{m}}\right)f\left(r_{j}\right)K\left(\frac{r_{j}-x}{h_{m}}\right)\right)dr_{i}dr_{j}\right)\\ ={\left(mh_{m}^{n}\right)}^{-2}\left(m\int f\left(r\right)K^{2}\left(\frac{r-x}{h_{m}}\right)dr+m\left(m-1\right){\left(\int f\left(r\right)K\left(\frac{r-x}{h_{m}}\right)dr\right)}^{2}\right)\\ ={\left(mh_{m}^{n}\right)}^{-2}\left(mh_{m}^{n}\int K^{2}\left(u\right)f\left(x+h_{m}u\right)du+m\left(m-1\right){\left(h_{m}^{n}\int f\left(x+h_{m}u\right)K\left(u\right)du\right)}^{2}\right). \end{array}$$From Equation (Equation 23),
(25)$$E\left[{\hat{f}}_{K}\left(x\right)\right]-f\left(x\right)=\frac{h_{m}^{2}}{2}\int u^{T}\left(\frac{\partial^{2}f\left(x+\theta h_{m}u\right)}{\partial x\partial x^{T}}\right)uK\left(u\right)du,$$
where θ represents a constant value between 0 and 1. According to Equations (Equation 23) and (Equation 24),
(26)$$E\left[{\hat{f}}_{K}^{2}\left(x\right)\right]-{\left(E\left[{\hat{f}}_{K}\left(x\right)\right]\right)}^{2}=\frac{\int K^{2}\left(u\right)f\left(x+h_{m}u\right)du}{mh_{m}^{n}}-\frac{{\left(\int K\left(u\right)f\left(x+h_{m}u\right)du\right)}^{2}}{m}.$$According to the Equations (Equation 25) and (Equation 26), the following equation holds.
(27)$$\begin{array}{cc} E{\left[{\hat{f}}_{K}\left(x\right)-f\left(x\right)\right]}^{2} & =E\left[{\hat{f}}_{K}^{2}\left(x\right)\right]-{\left(E\left[{\hat{f}}_{K}\left(x\right)\right]\right)}^{2}+{\left(E\left[{\hat{f}}_{K}\left(x\right)\right]-f\left(x\right)\right)}^{2}\\ & =\frac{\int K^{2}\left(u\right)f\left(x+h_{m}u\right)du}{mh_{m}^{n}}-\frac{{\left(\int K\left(u\right)f\left(x+h_{m}u\right)du\right)}^{2}}{m}\\ & +{\left(\frac{1}{2}h_{m}^{2}\int u^{T}\left(\frac{\partial^{2}f\left(x+\theta h_{m}u\right)}{\partial x\partial x^{T}}\right)uK\left(u\right)du\right)}^{2} \end{array}$$To facilitate the subsequent reasoning, the following theorem is given.**Theorem** **2.**
*For any matrix*
**Φ**
*, $K\left(\cdot \right)$ is a kernel density function with symmetric form; then,*
(28)$$\int x^{T}\Phi xK\left(x\right)dx=\frac{tr\left(\Phi \right)}{n}\int x^{T}xK\left(x\right)dx.$$
**Proof.** If the odd function $g\left(x\right)$ is integrable on R, there must be $\int_{-\infty }^{\infty }g\left(x\right)dx=0$. Similarly, it can be verified that the kernel function $K\left(\cdot \right)$ with a symmetric form satisfies
(29)$$\int \ldots \int \sum_{i\neq j}\Phi_{ij}x_{i}x_{j}K\left(x\right)dx_{1}\ldots dx_{n}=0.$$Then,
(30)$$\begin{array}{cc} \; & \int x^{T}\Phi xK\left(x\right)dx=\int \ldots \int x^{T}\Phi xK\left(x\right)dx_{1}\ldots dx_{n}\\ \; & =\int \ldots \int \sum_{i}\Phi_{ii}x_{i}^{2}K\left(x\right)dx_{1}\ldots dx_{n}+\int \ldots \int \sum_{i\neq j}\Phi_{ij}x_{i}x_{j}K\left(x\right)dx_{1}\ldots dx_{n}\\ \; & =\mathrm{tr}\left(\Phi \right)\int \ldots \int x_{1}^{2}K\left(x\right)dx_{1}\ldots dx_{n}\\ \; & =\frac{tr\left(\Phi \right)}{n}\int \ldots \int x^{T}xK\left(x\right)dx_{1}\ldots dx_{n}\\ \; & =\frac{tr\left(\Phi \right)}{n}\int x^{T}xK\left(x\right)dx. \end{array}$$Thus, the Theorem 2 is proved. □For any unit length vector $u\in R^{n}$, the Taylor expansion can be used to obtain
(31)$$\left\{\begin{array}{cc} \; & f\left(x+h_{m}u\right)=f\left(x\right)+h_{m}u^{T}\nabla \left(f\left(x\right)\right)+o\left(h_{m}\right)\\ \; & \frac{\partial^{2}f\left(x+\theta h_{m}u\right)}{\partial x_{i}\partial x_{j}}=\frac{\partial^{2}f\left(x\right)}{\partial x_{i}\partial x_{j}}+\theta h_{m}u^{T}\nabla \left(\frac{\partial^{2}f\left(x\right)}{\partial x_{i}\partial x_{j}}\right)+o\left(h_{m}\right) \end{array}\right.$$If the bandwidth $h_{m}$ satisfies the condition
(32)$$\left\{\begin{array}{cc} \; & \lim_{m\to \infty }\left(h_{m}\right)=0,\\ \; & \lim_{m\to \infty }\left(\frac{1}{mh_{m}^{n}}\right)=0, \end{array}\right.$$Then, from Equations (Equation 22)–(Equation 32), we get that
(33)$$E{\left[{\hat{f}}_{K}\left(x\right)-f\left(x\right)\right]}^{2}=\frac{c_{K}f\left(x\right)}{mh_{m}^{n}}+o\left(\frac{1}{mh_{m}^{n}}\right)+\frac{h_{m}^{4}d_{K}^{2}}{4n^{2}}{\left(\mathrm{tr}\left(\frac{\partial^{2}f\left(x\right)}{\partial x\partial x^{T}}\right)\right)}^{2}+o\left(h_{m}^{4}\right).$$In fact,
(34)$$\begin{array}{cc} & \begin{array}{cc} E{\left[{\hat{f}}_{K}\left(x\right)-f\left(x\right)\right]}^{2} & =\frac{\int K^{2}\left(u\right)f\left(x+h_{m}u\right)du}{mh_{m}^{n}}-\frac{{\left(\int K\left(u\right)f\left(x+h_{m}u\right)du\right)}^{2}}{m}\\ & +{\left(\frac{h_{m}^{2}}{2}\int u^{T}\left(\frac{\partial^{2}f\left(x+\theta h_{m}u\right)}{\partial x\partial x^{T}}\right)uK\left(u\right)du\right)}^{2} \end{array}\\ & =\frac{c_{K}f\left(x\right)}{mh_{m}^{n}}+o\left(\frac{1}{mh_{m}^{n}}\right)-\frac{f{\left(x\right)}^{2}}{m}+o\left(\frac{1}{m}\right)+{\left(\frac{h_{m}^{2}}{2n}\mathrm{tr}\left(\frac{\partial^{2}f\left(x+\theta h_{m}u\right)}{\partial x\partial x^{T}}\right)\int u^{T}uK\left(u\right)du\right)}^{2}\\ & =\frac{c_{K}f\left(x\right)}{mh_{m}^{n}}+o\left(\frac{1}{mh_{m}^{n}}\right)+{\left(\frac{h_{m}^{2}}{2n}\mathrm{tr}\left(\frac{\partial^{2}f\left(x\right)}{\partial x\partial x^{T}}\right)d_{K}+o\left(h_{m}^{2}\right)\right)}^{2}\\ & =\frac{c_{K}f\left(x\right)}{mh_{m}^{n}}+o\left(\frac{1}{mh_{m}^{n}}\right)+\frac{h_{m}^{4}d_{K}^{2}}{4n^{2}}{\left(\mathrm{tr}\left(\frac{\partial^{2}f\left(x\right)}{\partial x\partial x^{T}}\right)\right)}^{2}+o\left(h_{m}^{4}\right). \end{array}$$Based on Equation (Equation 33), if $\mathrm{tr}\left(\frac{\partial^{2}f\left(x\right)}{\partial x\partial x^{T}}\right)$ is integrable, there is
(35)$$\begin{array}{cc} \mathrm{MISE}\left({\hat{f}}_{K}\left(x\right)\right) & =\int \left(\frac{c_{K}f\left(x\right)}{mh_{m}^{n}}+\frac{h_{m}^{4}}{4n^{2}}{\left(d_{K}\mathrm{tr}\left(\frac{\partial^{2}f\left(x\right)}{\partial x\partial x^{T}}\right)\right)}^{2}\right)dx+o\left(\frac{1}{mh_{m}^{n}}\right)+o\left(h_{m}\right)\\ & =\frac{c_{K}}{mh_{m}^{n}}+\frac{1}{4n^{2}}h_{m}^{4}d_{K}^{2}\int \mathrm{tr}{\left(\frac{\partial^{2}f\left(x\right)}{\partial x\partial x^{T}}\right)}^{2}dx+o\left(\frac{1}{mh_{m}^{n}}\right)+o\left(h_{m}\right). \end{array}$$When $\mathrm{MISE}\left({\hat{f}}_{K}\left(\cdot \right)\right)$ is the smallest, the derivative of Equation (Equation 35) with respect to $h_{m}$ is 0, which means
(36)$$\frac{\partial MISE\left({\hat{f}}_{K}\left(x\right)\right)}{\partial h_{m}}=0.$$Thus, the optimal bandwidth $h_{m}$ in Theorem 1 is obtained as
(37)hm=mdK2n3cK∫tr∂2fx∂x∂xT2dx−11n+4n+4. □

**Remark** **2.**
*When the number of samples m is determined, the appropriate bandwidth $h_{m}$ can be selected using Equation (Equation 20) to construct the KDE, which can better fit the sample distribution. In Equation (Equation 20), the influence of the kernel function on bandwidth selection is on $c_{K}$ and $d_{K}$, which are almost the same under different kernel function selection, and they have a slight effect on the final bandwidth selection.*


### 3.2. Optimal Bandwidth Algorithm

The optimal bandwidth formula is given by Equation (Equation 20). However, $f\left(x\right)$ is unknown in Equation (Equation 20), and therefore, $\int \mathrm{tr}\left(\frac{\partial^{2}f\left(x\right)}{\partial x\partial x^{T}}\right)dx$ is also unknown. An approximate value of the bandwidth parameter $h_{m}$ can be obtained by replacing $f\left(x\right)$ with ${\hat{f}}_{K}\left(x\right)$ in Equation (Equation 16). Furthermore, an iterative algorithm can be used to calculate a more accurate bandwidth parameter. Theorem 3 shows that the algorithm is convergent.

**Theorem** **3.**
*For any n-dimensional probability density function $f\left(\cdot \right)$ and Gaussian kernel function $K\left(\cdot \right)$, if ${\hat{f}}_{K}\left(\cdot \right)$ in Equation (Equation 16) is used to estimate $f\left(\cdot \right)$, then the iterative calculation formula of $h_{m}$ is obtained as*
(38)hm,k+1=mdK2n3cK∫tr∂2f^Kx,hm,k∂x∂xT2dx−11n+4n+4

*and it is convergent, where $h_{m,k}$ is the value of $h_{m}$ during the k−th iteration.*


**Proof.** For a particular Gaussian kernel function
(39)Ku=2π−nn22e−uTuuTu22$d_{K}$ is a $\chi^{2}$ distribution with degree of freedom *n*, and the expectation is equal to the degree of freedom.
(40)$$d_{K}=\int u^{T}uK\left(u\right)du=n$$In addition,
(41)$$c_{K}=\int K^{2}\left(u\right)du=\int {\left(2\pi \right)}^{-n}e^{-u^{T}u}du={\left(2\sqrt{\pi }\right)}^{-n}.$$Substituting Equations (Equation 39)–(Equation 40) into Equation (Equation 20) and substituting ${\hat{f}}_{K}\left(x\right)$ in Equation (Equation 16) for $f\left(x\right)$, the iterative form of calculating $h_{m}$ is obtained as
(42)hm,k+1=nm11n+4n+42π−nnn+4n+4∫tr∂2f^Kx∂x∂xT2dx−11n+4n+4=mnhm,k2n11n+4n+42π−nnn+4n+4∫tr∂2∂x∂xT∑i=1mKri−xhm,k2dx−11n+4n+4To facilitate the subsequent reasoning, the following lemma is given as**Lemma** **1.**
*For any function $f_{1},f_{2},\ldots ,f_{n}$, inequality*
(43)$$\int {\left(f_{1}+f_{2}+\ldots +f_{n}\right)}^{2}dx\leq \int n\left(f_{1}^{2}+f_{2}^{2}+\ldots +f_{n}^{2}\right)dx.$$

*If and only if $f_{1}\left(x\right)=f_{2}\left(x\right)=\ldots =f_{n}\left(x\right)$ holds almost everywhere.*
**Proof.** In fact, for any function $f_{1},f_{2},\ldots ,f_{n}$, there are
(44)$$0\leq {\left(f_{1}\left(x\right)+f_{2}\left(x\right)+\ldots +f_{n}\left(x\right)\right)}^{2}\leq n\left(f_{1}{\left(x\right)}^{2}+f_{2}{\left(x\right)}^{2}+\ldots +f_{n}{\left(x\right)}^{2}\right).$$Thus, the two sides of Equation (Equation 44) are integrated as
(45)$$\int {\left(f_{1}+f_{2}+\ldots +f_{n}\right)}^{2}dx\leq \int n\left(f_{1}^{2}+f_{2}^{2}+\ldots +f_{n}^{2}\right)dx.$$It is obvious that the sign of Equation (Equation 43) holds the condition that $f_{1}\left(x\right)=f_{2}\left(x\right)=\ldots =f_{n}\left(x\right)$ is almost everywhere. □Because the second derivative of Equation (Equation 39) with respect to $x_{i}$ is
(46)∂2∂xi∂xiKx=2π−nn22e−xTxxTx22xi2−1.In addition,
(47)∫∂2∂xj∂xjKri−xhm,k2dx=∫∂2∂xj∂xj2π−nn22e−ri−xTri−xri−xTri−x2hm,k22hm,k22dx=342π−nhm,kn−4.From Lemma 1 and Equation (Equation 47)
(48)$$\begin{array}{cc} \int tr{\left(\frac{\partial^{2}}{\partial x\partial x^{T}}\left(\sum_{i=1}^{m}K\left(\frac{r_{i}-x}{h_{m,k}}\right)\right)\right)}^{2}dx & \leq \int nm\sum_{i,j}{\left(\frac{\partial^{2}}{\partial x_{j}\partial x_{j}}\left(K\left(\frac{r_{i}-x}{h_{m,k}}\right)\right)\right)}^{2}dx\\ & =\frac{3}{4}{\left(nm\right)}^{2}{\left(2\sqrt{\pi }\right)}^{-n}h_{m,k}^{n-4}. \end{array}$$When $h_{m,k}$ is sufficiently large, we can assume that $K\left(\frac{r_{i}-x}{h_{m,k}}\right)$ is almost the same everywhere, i.e., the equal sign in Equation (Equation 48) is tenable.
(49)hm,k+1=mnhm,k2n2π11n+4n+434nm22π−nhm,kn−4−11n+4n+4=hm,k34nm−11n+4n+4<hm,kWhen $h_{m,k}$ is large, the iterative process decreases. Because $h_{m,k}$ has a lower bound, the algorithm converges. □

In summary, the KDE method based on optimal bandwidth is given (See Algorithm 1), and the flowchart of the KDE method is shown in Figure 1.

**Algorithm 1:** Kernel density estimation (KDE) method based on optimal bandwidth.

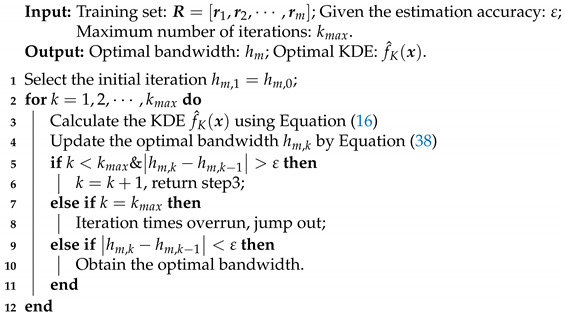



## 4. Fault Detection Method Based on JS Divergence Distribution

In Section 3, we construct a multidimensional KDE method based on the optimal bandwidth; this method can accurately describe the density distribution of multidimensional data. JS divergence is used to measure the distribution difference, and thus, it can highlight the difference in the statistical characteristics of different mode data.

### 4.1. Mode Difference Index

In Section 3, the probability density estimation of multidimensional data is obtained using the kernel function method, and the optimal bandwidth formula is derived. When the system fails, the state of the system will inevitably change, and the statistical characteristics of the system output will also change, thereby leading to significant changes in the density distribution of the observed data. For two groups of the sample window data R and Z, the cross entropy $H\left(R,Z\right)$ can be used to measure the distribution difference of R and Z.
(50)$$H\left(R,Z\right)=\int -{\hat{f}}_{K,Z}\left(x\right)\log\left({\hat{f}}_{K,R}\left(x\right)\right)dx,$$
where ${\hat{f}}_{K,R},{\hat{f}}_{K,Z}$ represents the optimal KDE of R and Z calculated using Equation (Equation 16).

$H\left(R,Z\right)$ does not satisfy the definition of distance because $H\left(R,Z\right)$ does not necessarily satisfy positive definiteness and symmetry; that is, $H\left(R,Z\right)<0$ or $H\left(R,Z\right)\neq H\left(R,Z\right)$.

The smaller the difference of distribution, the smaller is $H\left(R,Z\right)$, which means that even $H\left(R,Z\right)<0$, and therefore, it is reasonable to use $H\left(R,Z\right)$ to measure the distribution difference of R and Z.However, the quantitative description of distribution difference must satisfy symmetry; otherwise, the exchange position and distribution difference will be different, which is difficult to accept.

The JS divergence $JS\;\left(R,Z\right)$ was used as a measure of the distribution difference between R and Z in reference Zhang et al. [19], Bruni et al. [20] as follows:(51)$$JS\left(R,Z\right)=\int \begin{array}{cc} {\hat{f}}_{K,R}\log\left({\hat{f}}_{K,R}\right) & +{\hat{f}}_{K,Z}\log\left({\hat{f}}_{K,Z}\right)\\ & -\left({\hat{f}}_{K,R}+{\hat{f}}_{K,Z}\right)\log\left(\left({\hat{f}}_{K,R}+{\hat{f}}_{K,Z}\right)/2\right) \end{array}dx.$$

It is easy to get that
(52)$$\left\{\begin{array}{cc} \; & JS\left(R,Z\right)\geq 0\\ \; & JS\left(R,Z\right)=JS\left(Z,R\right) \end{array}\right.$$

In this paper, Equation (Equation 52) is used to measure the distribution difference between testing data Z and training data R for realizing fault detection and isolation.

### 4.2. Mode Discrimination Method

If the training data has *q* patterns $\left\{R_{1},R_{2},\ldots ,R_{q}\right\}$, the JS divergence set
$$\left\{JS\left(Z,R_{1}\right),JS\left(Z,R_{2}\right),\ldots ,JS\left(Z,R_{q}\right)\right\}$$
between the testing data Z and different modes R can be calculated using Equation (Equation 51).

If $i_{0}$ is the schema tag corresponding to the minimum JS divergence, it means that
(53)$$i_{0}=\arg\min\left\{JS\left(Z,R_{1}\right),JS\left(Z,R_{2}\right),\ldots ,JS\left(Z,R_{q}\right)\right\}.$$

It is reasonable to assume that testing data Z and training data $R_{i_{0}}$ belong to the same mode. However, for a new failure mode that may be unknown in the application, Equation (Equation 50) evaluates the testing data Z as the known failure mode of type $i_{0}$, which is obviously unreasonable.

If $JS\left(Z,R_{i_{0}}\right)$ is too large, we believe that testing data Z comes from an unknown new failure mode; its label is q+1. However, the method to obtain the threshold $JS_{\mathrm{high}}$ of $JS\left(Z,R_{i_{0}}\right)$ is a problem that should be investigated. A method to determine $JS_{\mathrm{high}}$ is provided below.

For the training data $R_{i_{0}}=\left[r_{1},r_{2},\ldots ,r_{m}\right]$ of the $i_{0}$ mode, the density estimation of the data set can be obtained using Equation (Equation 16).
(54)$${\hat{f}}_{K,R}\left(x\right)=\frac{1}{m{\left(h_{m}\right)}^{n}}\sum_{i=1}^{m}K\left(\frac{r_{i}-x}{h_{m}}\right)$$

In addition, if the length of the sampling window is fixed as $p\left(p<m\right)$, the new sampling data is $R^{\left(j\right)}=\left[r_{j},r_{j+1},\ldots ,r_{j+p}\right]\subset R_{i_{0}},j=1,2,\ldots ,m-p$ by sliding the sampling window. For each $R^{\left(j\right)}$, the density of the dataset can be estimated as
(55)$${\hat{f}}_{K,R^{\left(j\right)}}\left(x\right)=\frac{1}{p{\left(h_{p}\right)}^{n}}\sum_{i=j}^{j+p}K\left(\frac{r_{i}-x}{h_{p}}\right).$$

Using Equation (Equation 52), the divergence between the training data R and the sample data $R^{\left(j\right)}$ can be obtained as
(56)$$\begin{array}{cc} JS_{j}\; & =JS\left(R,R^{\left(j\right)}\right)\\ & =H\left(\left({\hat{f}}_{K,R}+{\hat{f}}_{K,R^{\left(j\right)}}\right),\left({\hat{f}}_{K,R}+{\hat{f}}_{K,R^{\left(j\right)}}\right)/2\right)-H\left({\hat{f}}_{K,R}\right)-H\left({\hat{f}}_{K,R^{\left(j\right)}}\right). \end{array}$$

Using Equation (Equation 55), we can obtain a series of JS divergence calculation value sets
$$\mathrm{JS}=\left\{JS_{1},JS_{2},\ldots ,JS_{m-p}\right\}.$$

We use this set to provide the estimation formula ${\hat{f}}_{JS}\left(x\right)$ of the density function $f_{JS}\left(x\right)$ of the JS divergence as
(57)$${\hat{f}}_{JS}\left(x\right)=\frac{1}{\left(m-p\right){\left(h_{m-p}\right)}^{n}}\sum_{j=1}^{m-p}K\left(\frac{JS_{j}-x}{h_{m-p}}\right).$$

If the significance level is α, the probability of ${\hat{f}}_{JS}\left(x\right)$ that exceeds the threshold $JS_{\mathrm{high}}$ is
(58)$$P\left\{\int_{0}^{JS_{\mathrm{high}}}{\hat{f}}_{JS}\left(x\right)dx.\right\}<\alpha$$

Because the distribution type of JS divergence is not a common random distribution, the quantile cannot be obtained by looking up the table; instead, it can only be obtained by numerical integration. If *h* is the step size, and
(59)$$\int_{h*\left(i-1\right)}^{+\infty }{\hat{f}}_{JS}\left(x\right)dx\leq \alpha \leq \int_{h*i}^{+\infty }{\hat{f}}_{JS}\left(x\right)dx,$$

It is reasonable to deduce that
(60)$$JS_{\mathrm{high}}=h*i.$$

The following fault detection and isolation criteria are constructed by Equation (Equation 58).

**Criterion** **1.**
*Suppose $i_{0}$ is the pattern label corresponding to the minimum JS divergence—see Equation (Equation 38)—the training data $R_{i_{0}}=\left[r_{1},r_{2},\ldots ,r_{m}\right]$ corresponding to the $i_{0}$ mode and the upper bound of JS divergence is $JS_{\mathrm{high}}$—see Equation (Equation 56). If the testing data $Z=\left[z_{1},z_{2},\ldots ,z_{l}\right]$ meet the requirements,*
(61)$$JS\left(Z,R_{i_{0}}\right)\leq JS_{\mathrm{high}}.$$

*The testing data Z and training data $R_{i_{0}}$ belong to the same failure mode; otherwise, the testing data Z are considered to originate from the unknown new failure mode, and their label is marked as q+1.*


In conclusion, the fault diagnosis method based on optimal bandwidth is provided (See Algorithm 2), and the corresponding fault diagnosis method flowchart is shown in Figure 2.

**Algorithm 2:** Fault Diagnosis
Method Based on Optimal KDE.

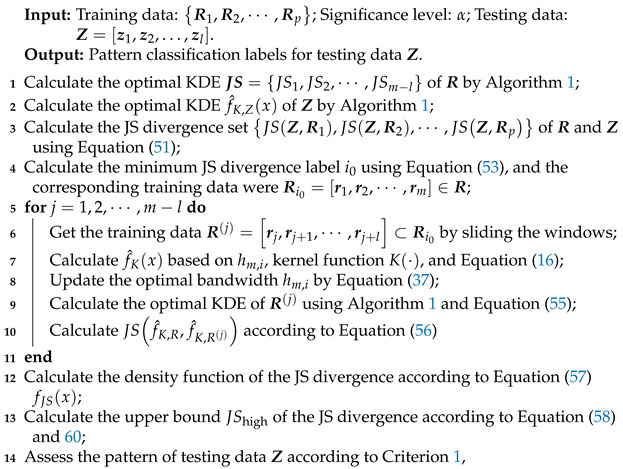



**Remark** **3.**
*Equations (Equation 54) and (Equation 55) show that the calculation result of JS divergence is directly related to the length of sampling data. Indeed, with the increase in the sampling data length, the density estimation obtained by Equation (Equation 54) can describe the distribution characteristics of samples more effectively, thereby significantly improving the accuracy of fault detection.*


## 5. Numerical Simulation

The bearing data from Case Western Reserve University Bearing Data Center were used as the diagnosis research object, and they have been considered as a case for many fault diagnosis, such as in references Smith and Randall [21], Lou and Loparo [22], Rai and Mohanty [23].

The sampling frequency of the motor data was 12 kHz, and 12 kHz is the default sampling frequency for Case Western Reserve University Bearing Data Center. The dataset contains four groups of sample data: normal data ($f_{0}$), 0.007 inch inner raceway fault data ($f_{1}$), 0.014 inch inner raceway fault data ($f_{2}$), and 0.014 inch outer raceway fault data($f_{3}$). Each group of data had two dimensions: the acceleration data of the drive end ($f_{i}-DE$) and the acceleration data of the fan end ($f_{i}-FE$). All the experiments were conducted on an Lenovo Ryzen 3700X CPU with 3.60 GHz processor, 16 GB RAM.

### 5.1. Data Preprocessing

The observed data in the process of the bearing operation show obvious periodicity, which needs to be eliminated. Taking normal data $f_{0}$ as an example, the main frequency in the observed signal can be obtained by fast Fourier transform (FFT), and the result of the FFT is shown in Figure 3.

Figure 3 indicates that the main frequency is approximately 1036 Hz, and thus, the basis function is constructed as
$$f\left(t\right)={\left[\begin{array}{ccc} 1 & \sin\left(1036\times 2\pi t\right) & \cos\left(1036\times 2\pi t\right) \end{array}\right]}^{T}.$$

The estimation of β calculated using Equation (Equation 7) is
$$\hat{\beta }=\left[\begin{array}{ccc} 0.0116 & -0.0158 & 0.0548\\ 0.0280 & 0.0326 & -0.0396 \end{array}\right].$$

Thus, the data after removing the intrinsic signal are shown in Figure 4, where Figure 4a represents the acceleration data of the drive end and Figure 4b represents the acceleration data of the fan end.

In the later fault detection process, the data of all modes are similar to the above operation, and the results are recorded as $f_{i}$.

### 5.2. Fault Detection Effect

#### 5.2.1. Norm Data and Known Fault

For the norm data $f_{0}$ and the known fault $f_{1},f_{2}$, the first 20,480 sample points are selected as the training set, which are recorded as $f_{i-\mathrm{train}}$. The last 81,920 sample points are taken as the testing set, which are recorded as $f_{i-\mathrm{test}}$. A total of 128 sample points are used as detection objects in each test. The training set data are shown in Figure 5, where Figure 5a,b represent data $f_{i-\mathrm{train}},i=1,2$ of the two dimensions, respectively.

Figure 5 shows that the bearing data have high frequency, and the fault does not change the observed mean value; however, it changes the dispersion characteristics or the correlation of data.

#### 5.2.2. Unknown Fault

The training data does not necessarily contain all types of patterns, and the detection of unknown faults is always a difficult problem. $f_{3}$ is used as an unknown fault for fault detection; the training set sample does not contain any information about $f_{3}$. The unknown fault data are shown in Figure 6, where in Figure 6a represents the acceleration data at the driving end and Figure 6b represents the acceleration data at the fan end.

Figure 6 shows that the data of unknown faults is close to the other two types of fault data. If the fault detection method is not sensitive, the detection rate will be reduced significantly.

#### 5.2.3. Detection Effect

The characteristics of bearing data make bearing fault detection extremely challenging. The input of the training set is $f_{0-\mathrm{train}}$, the estimation accuracy is $\varepsilon =10^{-4}$, and the maximum number of iterations is $k_{max}=100$, according to Algorithm 1, the optimal bandwidth is
$$h_{m}=0.0445.$$

The KDE of the training set is obtained by Equation (Equation 15), and the results are shown in Figure 7, where Figure 7a,c,e represent the two-dimensional frequency histograms of the training data $f_{i-\mathrm{train}},i=0,1,2$, and Figure 7b,d,f represent the two-dimensional KDE of the training data $f_{i-\mathrm{train}},i=0,1,2$.

Figure 7 further shows that the bearing fault changes the dispersion characteristics and data correlation. Meanwhile, Figure 7 shows that the KDE of the training data obtained by Equation (Equation 15) is in good agreement with the data distribution of the training data, and therefore, this method can really describe the distribution of multidimensional data.

The JS divergence of the training data and KDE of the distribution are obtained by Equations (Equation 51) and (Equation 58); the results are shown in Figure 8.

When the significance level is α=95%, the detection thresholds of the training set, which are calculated using Equation (Equation 58), are
$$\left\{\begin{array}{c} f_{0}:JS_{\mathrm{high}}<0.1375\\ f_{1}:JS_{\mathrm{high}}<0.0995\\ f_{2}:JS_{\mathrm{high}}<0.1225 \end{array}\right.$$

Thus, the detection results of using JS divergence methods on the testing data are shown in Figure 9. If the detection points fall within the threshold, the data set to be detected is in the same pattern; otherwise, the data have different patterns.

Furthermore, detection rates using different methods are shown in Table 1.

For the known fault, Table 1 indicates that the bearing fault identification based on multidimensional KDE and JS divergence achieves better results compared to those obtained using the $T^{2}$ statistics detection methods in the testing data. The detection rate of normal data $f_{0}$ increases from 95.08% to 97.03%, the detection rate of fault data $f_{1}$ increases from 81.33% to 95.81%, and the detection rate of fault data $f_{2}$ increases from 70.69% to 95.36%. Meanwhile, compared with the cross-entropy methods, the detection rate of normal data $f_{0}$ increased from 96.95% to 97.03%; of fault data $f_{1}$ increased from 94.41% to 95.81%; and of fault data $f_{2}$ increased from 94.19% to 95.36%.

For the unknown fault $f_{3}$, Table 1 shows that the $T^{2}$ statistics detection method cannot detect the unknown faults. The method using cross entropy as a measure can only detect unknown faults with a detection rate of 53.16%, which is not obvious. The JS divergence method constructed in this study can identify the unknown fault accurately, and the detection rate reaches 69.49%. This is because JS divergence is more accurate at measuring the differences between distributions.

### 5.3. Influence of Window Width on Fault Diagnosis

The fault diagnosis effect is related to the data window width; therefore, the fault diagnosis effect under different window widths is investigated. The results are shown in Figure 10.

Figure 10 indicates that, with the increase in the detection window, the detection performance of the proposed method for the known fault detection first rises, and then, it tends to be stable. This is because when the length of the detection window increases to a certain extent, the data to be detected already contains sufficient information. Meanwhile, if the detection window continues to increase, the contribution rate to the improvement of the fault detection rate is not large. Meanwhile, for unknown faults, the detection rate increases rapidly with the length of the detection window because the longer the detection window, the higher the amount of information contained in the data to be detected, and the better is the difference characterized between the fault and the known fault.

## 6. Conclusions

In this study, a method of bearing fault detection and identification was constructed using multidimensional KDE and JS divergence. The distribution characteristics of JS divergence between the sample density distribution and population density distribution were derived using the sliding sampling window method. Thus, the threshold of fault detection was provided, and therefore, different faults, especially unknown faults, could be identified. The theory showed that the multidimensional KDE method could reduce information loss caused by processing each dimension; the JS divergence is more accurate than the traditional cross entropy to measure the difference in density distribution. The experimental results verified the above conclusions.

For a known fault, the detection effect of this method was obviously better than that of the traditional method, and it also had a certain degree of improvement compared with the cross-entropy method. Second, for unknown faults, the traditional method could not detect the distribution difference accurately, while the detection effect of the proposed method was significantly improved.

Furthermore, the detection effect of this method depends on the window width. The detection effect improved with a growth in the detection window. In this paper, under the condition of a given window width, the estimation formula for the optimal bandwidth of a multidimensional KDE was provided. The experimental results showed that the formula was applicable to any mode of data, and therefore, it had a certain universality.

However, this study has certain limitations. Firstly, although the calculation formula of multidimensional KDE is given in this study, the computational complexity will increase when the dimension is large, which may restrict the further application of the method. Secondly, the calculation of JS divergence is time consuming, which is not conducive to rapid fault diagnosis.

In future research, we can try to use the PCA dimension reduction method to solve the computational complexity caused by very large dimension, and optimize the algorithm flow of JS divergence to expedite the calculation. In the latest study Ginzarly et al. [24], prognosis of the vehicle’s electrical machine is treated using a hidden Markov model after modeling the electrical machine using the finite element method. Therefore, we will try to combine this method in future work and apply it to the fault detection of other systems.

## Figures and Tables

**Figure 1 entropy-23-00266-f001:**
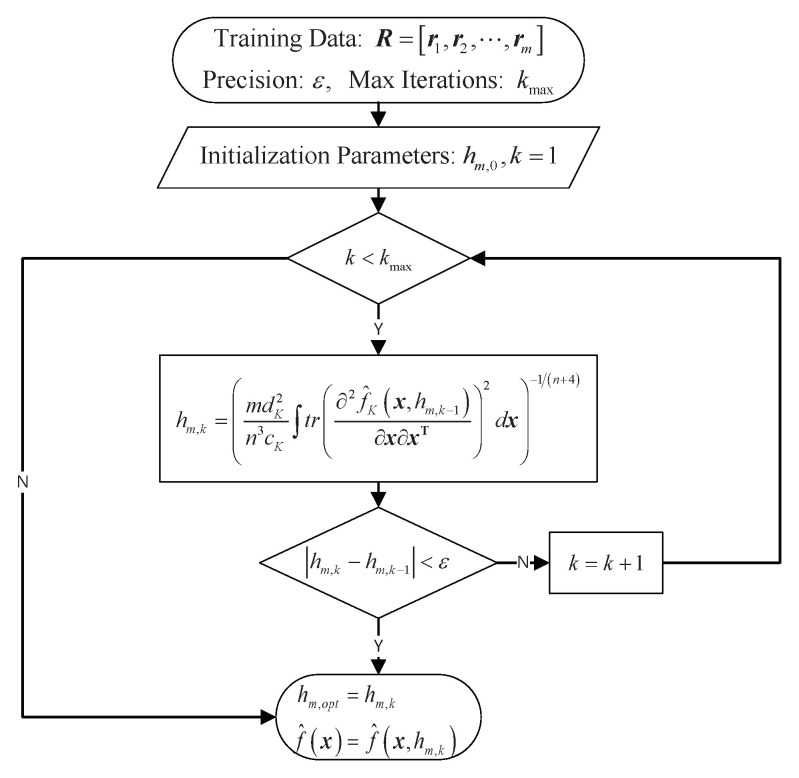
Flowchart of KDE method based on optimal bandwidth.

**Figure 2 entropy-23-00266-f002:**
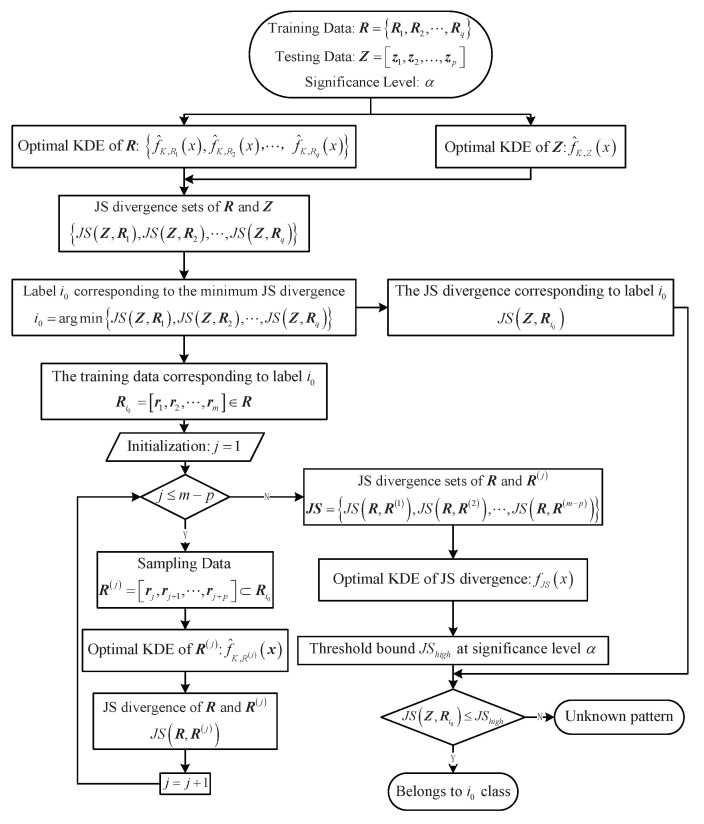
Flowchart of fault diagnosis method based on optimal KDE.

**Figure 3 entropy-23-00266-f003:**
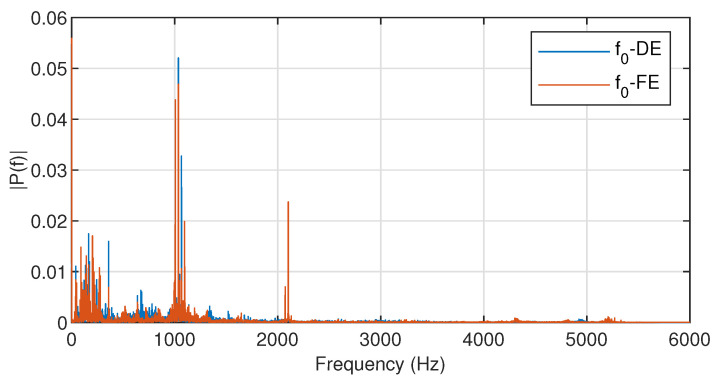
Single-sided amplitude spectrum of $f_{0}$.

**Figure 4 entropy-23-00266-f004:**
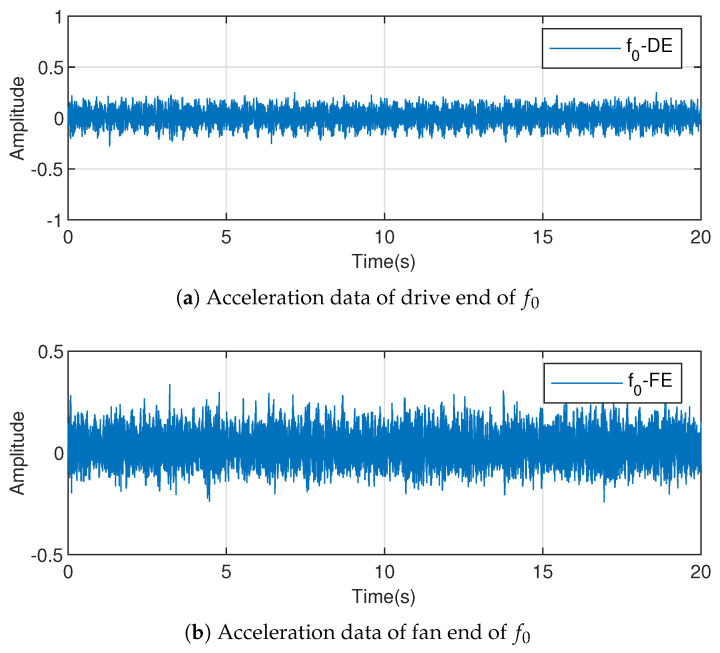
Preprocessed data to remove trends by fast Fourier transform (FFT).

**Figure 5 entropy-23-00266-f005:**
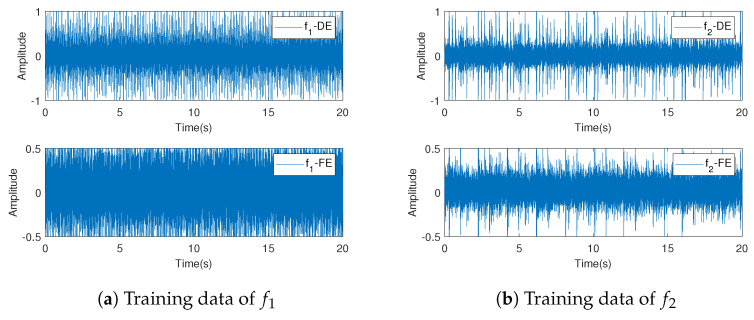
Training data $f_{1},f_{2}$ after being preprocessed.

**Figure 6 entropy-23-00266-f006:**
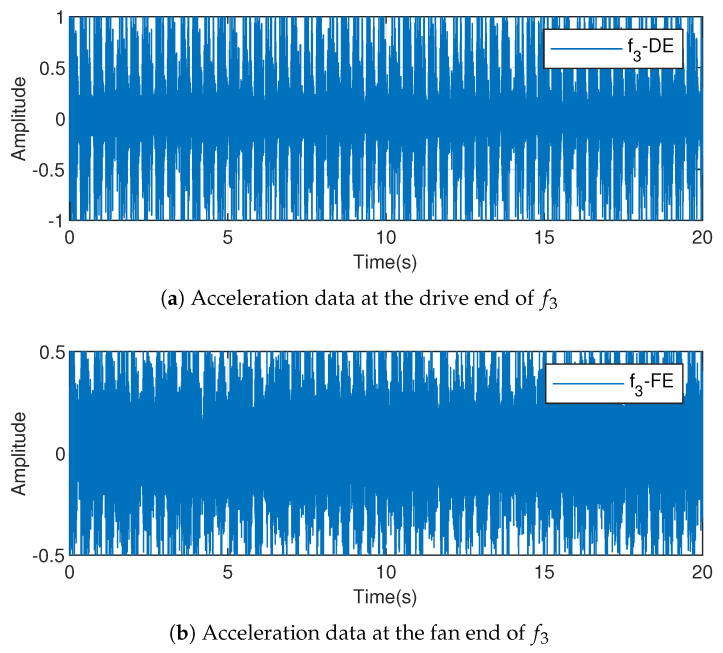
Training data $f_{3}$ after preprocessed.

**Figure 7 entropy-23-00266-f007:**
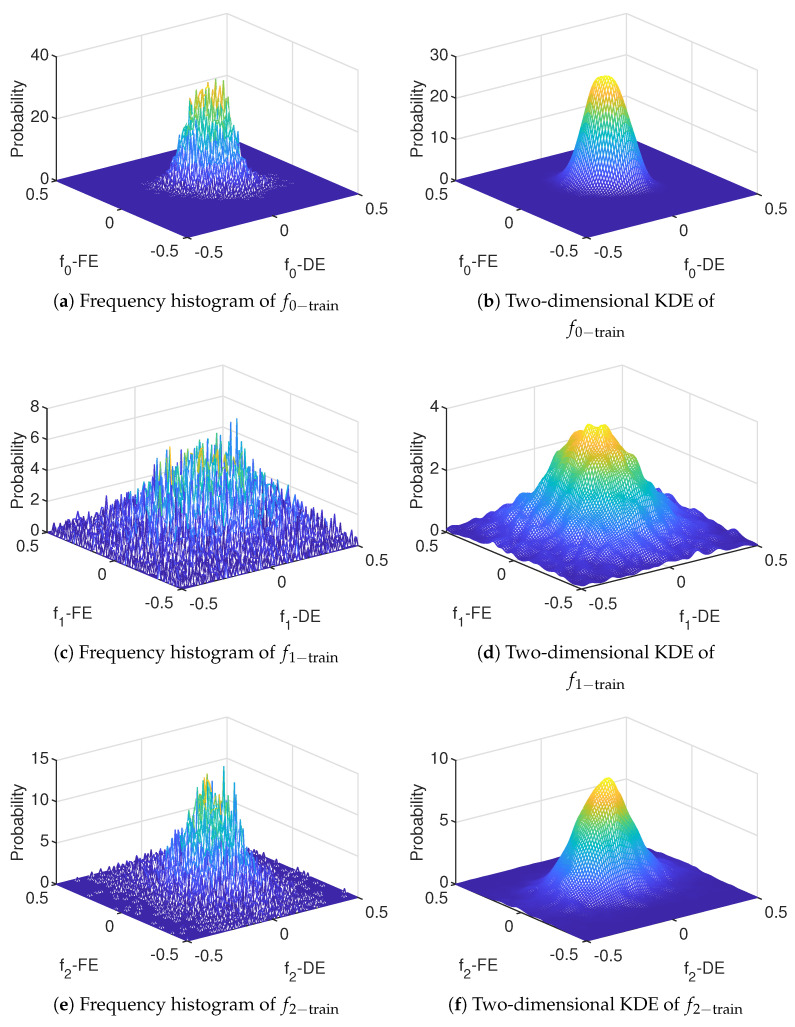
Training data after being preprocessed.

**Figure 8 entropy-23-00266-f008:**
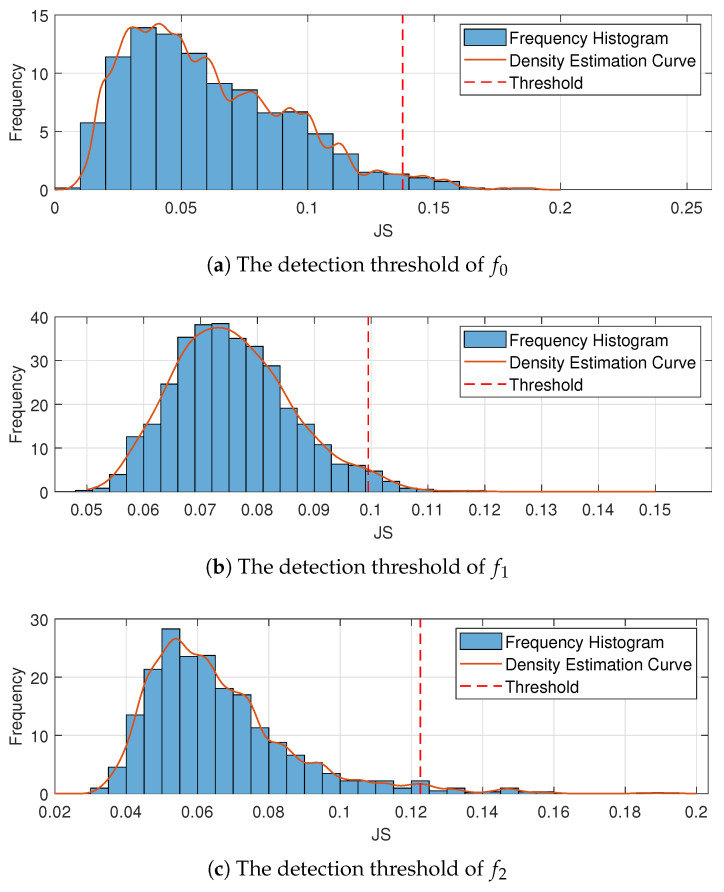
The results of detection threshold.

**Figure 9 entropy-23-00266-f009:**
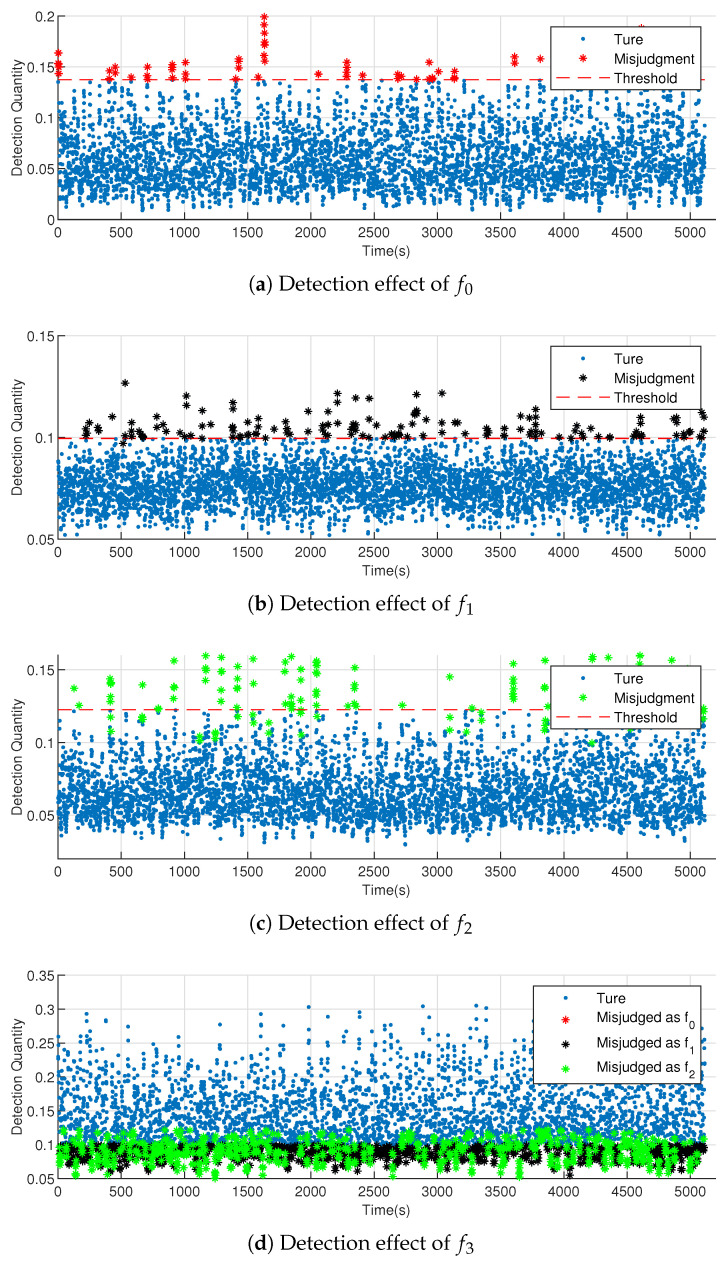
Fault detection effect using JS divergence as index.

**Figure 10 entropy-23-00266-f010:**
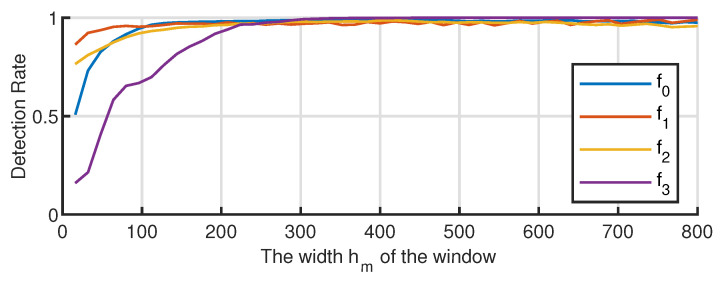
Fault diagnosis effect under different window width $h_{m}$.

**Table 1 entropy-23-00266-t001:** Detection rate of normal and different failure modes using different methods.

Method	$T^{2}$ Statistics Detection	Cross Entropy	JS Divergence
Normal mode $f_{0}$	95.80%	96.95%	97.03%
Known fault $f_{1}$	83.47%	94.41%	95.81%
Known fault $f_{2}$	78.11%	94.19%	95.36%
Unknown fault $f_{3}$	\	53.16%	69.49%

## Data Availability

Not applicable.

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
