# Peer review of "Fault Detection Based on Multi-Dimensional KDE and Jensen–Shannon Divergence"

_entropy, 2021, doi:10.3390/e23030266_

Round 1
Reviewer 1 Report
A new fault diagnosis method based on optimal bandwidth kernel density estimation (KDE) and Jensen–Shannon (JS) divergence distribution for improved fault detection performance is proposed. The method is validated by using the bearing data from Case Western Reserve University Bearing Data Center. Although some improvements are shown, several aspects have to be further discussed in order to show contribution and applicability.
Lines 51-56: Three main problems are discussed; however, it is not clear how the results obtained from the bearing data demonstrate that your proposal overcomes these problems.
Line 71: Differences with references 12, 13, 14, and 15 are not clear, since they also propose methods based on KDE.
Equations presented in section 1 need references. Are they yours?
Line 114: It is said that the basis function is not the focus of your paper; yet, it is very important for the results. Therefore, the impacts of using different functions or ways to select a proper function for different scenarios have to be discussed.
Line 147: Is yours the information presented from line 147-158 (more than two pages)? Information that is not yours can obscure your contributions. Please clarify it. The same for many other equations into the article.
Line 217: It is not clear why the data of the motor is 12kHz.
For figure 4, include the FFT in order to show the reduction of the fundamental component.
What is the impact of the results if another fundamental component (e.g., 2100 approximately) is removed after removing the frequency component of 1036? Please widely discuss this since the remaining fundamental component will always represent periodicity,
Please use the same y-scale in figure 5 to compare
Include all the parameters to reproduce algorithm 1 and obtain hm=0.0445. Also, several equations are used in section 4 but the used parameters are not mentioned at all.
Improve font size in Figures.
Computational burden is discussed in a qualitative way, quantitative results are needed. Include the software and hardware used for your results.
How can a user determine the window width for different applications? Also, parameters used in the algorithms are different for different applications, how can a user deal with that condition?
Also, recommendations or examples for other applications have to be included.
Please review, discuss and include entropy manuscripts to show the congruence with the current Entropy scope
Reviewer 2 Report
The bearing fault detection is a well known issue on any industrial application. Standard diagnosis involve vibration analysis by means of FFT decomposition. This approach is used by commercial devices from SKF, Brüel Kjaer, GE.
The presented methodology based on Kernel Density Estimation and Jensen-Shanon divergence distribution is interesting and well presented on the manuscript. The methodology is adequately presented.
The results are adequately presented. Nevertheless, authors compare the proposed methodology with statistical analysis. Authors should consider spectral decomposition analysis and Power spectral density on bands introduced on
"Vibration Analysis of Rolling Element Bearings Defects", Journal of Applied Research and Technology Volume 12, Issue 3, June 2014, Pages 384-395
Authors should provide more details on the experimental set to support the performance specially on the previously introduced challenges of
Incomplete Signal or Incomplete Data Set.
Reviewer 3 Report
In this paper, the authors propose a fault detection approach based on multi-dimensional KDE and JS divergence.
Comments
The manuscript is written clearly for readers to understand easily. And, the contributions are clearly identified both at the theoretical and practical level.
In this paper, the authors aim to detect incipient faults (small deviation of the system) which is a very interesting problem. The results show the efficiency of the proposed approach and limitations.
I recommend to the authors:
- To discuss in the introduction the link with Hidden Markov Model
- To extend, in a future work as perspectives, their approach from detection to prognosis
For these recommendations, the authors can read the following paper:
- From Modeling to Failure Prognosis of Permanent Magnet Synchronous Machine, 10 (2), p. 691, Applied Sciences, 2020.
Round 2
Reviewer 1 Report
All the comments and suggestions have been properly addressed. This Reviewer recommends the manuscript acceptance.
Reviewer 2 Report
The manuscript has been improved according reviewers comments.
Is ready to be publisehd